# Colistin Use in Patients with Extreme Renal Function: From Dialysis to Augmented Clearance

**DOI:** 10.3390/medicina55020033

**Published:** 2019-01-31

**Authors:** Aleksandra Aitullina, Angelika Krūmiņa, Šimons Svirskis, Santa Purviņa

**Affiliations:** 1Department of Pharmacology, Riga Stradins University, Pilsonu Street 13, LV-1002 Riga, Latvia; santa.purvina@rsu.lv; 2Department of Infectology and Dermatology, Riga Stradins University, Linezera Street 3, LV-1006 Riga, Latvia; angelika.krumina@rsu.lv; 3Institute of Microbiology and Virology, Riga Stradins University, Ratsupites Street 5, LV-1067 Riga, Latvia; simons.svirskis@rsu.lv

**Keywords:** colistin, augmented renal clearance, renal replacement therapy, *Acinetobacter baumannii*

## Abstract

*Background and objectives:* Colistin is used for the treatment of multidrug-resistant (MDR) Gram-negative bacteria infection in critically ill patients. It is recommended to adjust the dose in cases of renal impairment but not in cases of augmented renal clearance (ARC). The aim of this study was to determine colistin use pattern in patients with different renal functional states. *Materials and Methods:* Adult patients admitted to intensive care units of single Latvian hospitals in the years 2015–2017 with an MDR Gram-negative bacterial infection and at least 72 h colistin therapy were included in this study. Data were collected retrospectively from medical notes. Colistin use pattern and outcomes were analyzed in patients with different renal function prior to colistin therapy: normal, ARC, impaired, and on renal replacement therapy (RRT). *Results:* 100 cases of colistin use met the inclusion criteria. The study group was heterogeneous, and patients had different renal function states prior to colistin therapy-from continuous RRT (18 cases) to ARC (16 cases). The standard colistin dose of 9 million units (MU) daily was the most common dose among the patients. In many cases (43%), colistin dose adjustment did not follow the recent recommendations of drug manufacturers-this was mainly in patients with renal impairment prior to colistin therapy. Eighteen cases of colistin acute kidney injury (AKI) were detected, mostly (10 cases) in patients with normal renal function and without ARC prior to colistin therapy. No patients with colistin AKI needed RRT. *Conclusions*: Colistin doses varied greatly among patients, and in patients with renal function impairment it was commonly not in accordance with the summary of product characteristics (SPC). Patients with ARC mostly received a standard colistin daily dose of 9 MU daily, but the cumulative dose had a tendency to be higher and duration of colistin therapy was longer in comparison with other patient groups. ARC’s role in the development of colistin nephrotoxicity is still unclear.

## 1. Introduction

Colistin or polymyxin E is a cationic lipopeptide antibiotic that is administered intravenously as a non-active pro-drug colistimethate sodium (CMS) that is hydrolyzed in vivo to become an active substance [1]. This drug was discovered more than 60 years ago, but was not used for many years due to the concern of nephrotoxicity and the availability of other less-toxic antibacterial agents. At present, colistin is considered as a drug of last choice for the treatment of multidrug-resistant (MDR) systemic Gram-negative infections in critically ill patients [2].

CMS is eliminated mainly by the kidney, including renal tubular secretion, and colistin is eliminated predominantly by non-renal pathways [3]. If CMS is cleared via the kidney too rapidly, the systemic bioavailability of colistin could be reduced and that could theoretically lead to ineffective antibacterial therapy [4]. The phenomenon of enhanced renal function is called augmented renal clearance (ARC). There is a hypothesis that the development of ARC is related to the systemic inflammatory response syndrome (SIRS) that is observed in patients without chronic organ damage and with trauma, pancreatitis, burns, autoimmune disorders, major surgical procedures, ischemia, or sepsis. The hemodynamic manifestations of SIRS include high cardiac output, and due to this, enhanced renal blood flow. ARC is usually defined as the enhanced elimination of solute through the kidney with renal clearance above the expected baseline. However, in this case, it is problematic to define an ”expected or normal“ baseline value. The most widely accepted definition is a glomerular filtration rate (GFR) above 130 mL/min, best measured by 8–24 h urine collection [5]. Due to the potential risk of under-dosing renally cleared antibiotics, therapeutic drug monitoring (TDM) for these drugs may be advisable in this group of patients [6].

Colistin is potentially nephrotoxic, but its renal toxicity is usually reversible [7]. Renal impairment on baseline is not a contraindication for colistin administration, but dose adjustment is considered. In the case of renal replacement therapy (RRT), there is no mechanism to return colistin from dialysate back to blood. For this reason, colistin can be efficiently cleared during hemodialysis (HD) and therefore the authors of the Garonzik et al. 2011 study suggest that dialysis is best conducted toward the end of a CMS dosage interval [8]. Several years ago, there were no clear recommendations about colistin dose adjustment in RRT [2]. At present, drug manufacturers suggest the use of a standard colistin dose of 9 million units (MU) per day in patients with normal renal function and on continuous RRT (CRRT). This dose is reduced in cases of renal impairment (i.e., to 5.5–7.5 MU/day when creatinine clearance (CrCl) <50–30 mL/min; to 4.5–5.5 MU/day when CrCl <30–10 mL/min; and to 3.5 MU/day when CrCl <10 mL/min). Patients on intermittent HD (IHD) should receive 2.25 MU/day in non-HD days and 3 MU/day after HD session in HD days [9]. In the case of CRRT, dosing recommendations could also depend on different factors, such as the filter type used for dialysis. There are data suggesting that some hyperadsorptive filters (e.g., AN69ST) could adsorb colistin, and thus higher doses might be needed (e.g., 13.5 instead of 9 MU/day) [10].

The aim of this study was to determine the colistin use pattern in patients with different renal functional states, including extreme renal functions, in a single clinical center in a Baltic country.

## 2. Material and Methods

The study setting was in intensive care units (ICUs) of a tertiary adult university hospital in Riga, Latvia. The inclusion criteria were: adult patients, admitted to the ICU in the period of the study (2015–2017), with bacteriologically documented carbapenem-resistant Gram-negative bacterial infection, and systemic colistin therapy that started in the ICU. The exclusion criterion was colistin use less than 72 h. Information about patients’ demographics, blood test results, clinical diagnoses, duration of hospitalization and outcomes, colistin dosage regimens and duration of therapy, as well as bacteriological test results and colistin minimal inhibitory concentratiion (MIC) were collected retrospectively from medical notes. The colistin MIC breakpoint for *Acinetobacter baumannii* and *Pseudomonas aeruginosa* is 2 mg/L according to the European Committee on Antimicrobial Susceptibility Testing (EUCAST) data (v. 8.0; 01.01.2018). Patients who received a second intravenous colistin course after a cured infectious episode were considered as two separates cases when analyzing colistin use pattern and nephrotoxicity. On the other hand, in the demonstration of demographics and clinical data of the study population, every patient was presented as a separate case regardless of the number of colistin courses administered during hospitalization.

To reduce the risk of underestimation of renal function in cases of ARC, the assessment of baseline renal function in non-dialyzed patients was based on the Chronic Kidney Disease Epidemiology Collaboration (CKD-EPI) formula with a cut-off value for ARC of 108 mL/min/1.73 m^2^ [11]. The patients were divided into four groups according to renal function state prior to colistin therapy: the normal renal function group (defined as having a GFR of more than 50 mL/min/1.73 m^2^ and less than 108 mL/min/1.73 m^2^), the ARC group (defined as having a GFR of more than 108 mL/min/1.73 m^2^), the renal function impairment group (defined as having GFR of less than 50 mL/min/1.73 m^2^ or on IHD), and the CRRT group.

Colistin nephrotoxicity or colistin-induced acute kidney injury (AKI) was defined according to Acute Kidney Injury Network criteria as increasing the baseline creatinine level more than 1.5 times over baseline after at least 48 h of colistin therapy. The first stage of AKI was considered in cases of serum creatinine level increasing 1.5–1.9 times over baseline, the second stage in cases of serum creatinine increasing 2.0–2.9 times over baseline, and the third stage in cases of serum creatinine increasing more than 3.0 times over baseline [12]. The return to normal renal function (GFR >50 mL/min) in patients with ARC was not considered as colistin AKI. If acute renal function decline was observed in the day of patient death, these cases were excluded from further colistin AKI analysis. The study was conducted in accordance with the Declaration of Helsinki, and the protocol was approved by the Ethics Committee (48/05.10.2017).

### Statistical Analysis

Continuous data were expressed in the form of medians and interquartile ranges (Q1; Q3). Categorical data were expressed as counts and percentages. Normal distribution was assessed using the Shapiro-Wilk test. Mann-Whitney U-tests, paired *t*-tests and Kruskal-Wallis tests were used for data comparison amongst study populations. A *p*-value equal to or less than 0.05 was considered as statistically significant. Statistical data analysis was performed by IBM SPSS, version 22 (Statistical Package for the Social Sciences, Oshkosh, WI, USA).

## 3. Results

Ninety-seven medical histories met the inclusion criteria. Three patients received colistin therapy twice with an interval between colistin courses of more than one month, so 100 cases of colistin use were analyzed in total. Most of the included patients were men (65, or 67% of all cases). The median age was 63 years. *A. baumannii* infection was the most common causative agent for systemic MDR Gram-negative bacterial infection among patients in the study (98%), and the median hospitalization days from bacteriological documentation of this infection was nine days. In all cases, MDR bacteria were sensitive to colistin (the colistin MIC range for detected cases was 0.125–0.5 mg/L). MDR Gram-negative infection was found mostly in trachea aspirate (78, or 80.4% of cases). In 17 (17.5%) cases, it was isolated from the blood sample. The main data about patient demographics, clinical diagnoses, and MDR nosocomial infection are summarized in Table 1.

The study group was heterogeneous, and patients had different renal function states prior to colistin therapy. In 43 cases, GFR was from 50–108 mL/min (normal renal function group), in 16 cases GFR was more than 108 mL/min (ARC group), in 23 cases GFR was less than 50 mL/min (renal impairment group), and 18 patients were on CRRT on the day of starting of colistin therapy.

The most commonly used colistin regimen in all patient groups was 9 million units (MU) as a loading dose (LD) followed by 3 MU every 8 h. Patients with renal function impairment had a statistically higher probability of receiving smaller LD or of not receiving LD at all in comparison with patients with normal renal function or ARC (*p* = 0.002). In addition, patients with renal impairment had higher incidence of colistin under-dosing and over-dosing in comparison with other patient groups (Pearson chi-sq. *p* = 0) (Figure 1).

Patients in the ARC group were younger than patients with normal renal function or renal impairment (*p* = 0.0003). They also received a higher cumulative colistin dose for a longer period of time, but these two parameters did not reach statistical significance (Table 2).

Patients in the ARC group rarely experienced a significant increase of serum creatinine during colistin use, and some cases renal function just returned to a normal renal functional state (Table 3).

Eighteen cases (22%) met colistin AKI criteria, and these renal impairment cases were mostly mild or moderate (i.e., increase in serum creatinine of up to 3-fold) and AKI was mostly observed in patients with GFR of 50–129 mL/min prior to colistin therapy (10 cases of 18) with the median onset (Q1; Q3) of AKI being 7.5 (4.5; 16) days. The median serum creatinine prior to colistin therapy, patient age, cumulative colistin dose, and colistin over-dosing were not statistically significant risk factors of colistin AKI. On the other hand, higher C-reactive protein (CRP) on the background was associated with higher risk of colistin-induced AKI (Table 4). No patients with colistin AKI needed RRT.

## 4. Discussion

Colistin has been used intravenously for many years as the optimal strategy for dosing, especially in patients with renal impairment. Nevertheless, the research is unclear and varies among different clinical settings and hospitals [13]. In addition, this study revealed a wide variety of colistin dosing in patients with similar renal function. For example, patients with a GFR of 20–50 mL/min in the hospital of study received a wide range of colistin doses daily (i.e., from 1 to 9 MU). Some patients in this study received decreased LD of colistin or did not receive LD at all. Mostly, it was observed in the patient group with decreased renal function or those on CRRT. However, a LD for colistin is usually always recommended as it allows the patient to achieve a steady state of colistin much faster [14]. In addition, in this study, a significant number of patients (43%) received colistin doses that were higher or lower than is recommended in the SPC, especially patients with renal impairment. This wide variability in colistin dosing could be explained by the lack of clear recommendations about dose adjustment in the case of renal impairment during the years of the study. The advice from manufacturers about colistin dose adjustment were more suitable for non-critically ill patients, and clinicians were guided by different handbooks and articles that provide different recommendations [15]. For example, some patients on RRT in one ICU within the study received an increased dose of colistin by 4.5 MU every 8 h. This dosing regimen was based on the study results of Honore et al. regarding colistin adsorption on some hemodialysis filters [10]. However, other ICU patents on RRT received a decreased colistin dose rather than having it increased. In the previous year, the SPC of colistin was updated with information about colistin dosing in critically ill patients with different renal functional states. This theoretically could decrease the risk of over-dosing or under-dosing on colistin in this setting [9].

All possible types of kidney function were detected among patients in the study, ranging from RRT to ARC. Assessment of renal function in critically ill patients is a major challenge, as the most accurate methods for the detection of GFR (e.g., the use of exogenous filtration markers such as radionuclide markers or iohexal) are usually not used in routine clinical practice [16]. Estimated GFR based on serum creatinine level has a lot of limitations (e.g., it can be artificially increased in the case of malnutrition). However, it still commonly guides the decisions of clinicians about dose adjustment of the drug in real clinical situations.

In the SPC there are no specific recommendations for colistin dosing in the case of ARC. However, the SPC suggest two possible daily doses (i.e., 9 and 12 MU daily), accompanied with the comment that the safety of higher doses is not yet established [9]. Dalfino et al. suggest increased colistin doses for patients with augmented clearance, such as 12 MU daily when creatinine clearance (CrCl) >130 mL/min and 9 MU daily when CrCl is 60–130 mL/min [17]. Patients with ARC in this study mostly received a standard colistin dose of 9 MU daily, but they had the tendency to receive longer colistin therapy with a higher cumulative dose in comparison to other patient groups. However, this difference did not reach statistical significance.

It is postulated that ARC could increase the clearance of many drugs. In case of continuous infusions of sedatives or vasopressors, the dose could easily be adjusted according to clinical response. However, the dose of antibiotics in this patient group is widely discussed in the literature, and no indicator of decreased effectiveness is available. In addition, under-dosing could potentially lead to therapy failure and the development of bacterial resistance [16]. It is still unknown if colistin dose adjustment is needed in the case of ARC. When used for killing bacteria, colistin’s action is concentration-dependent rather than time-dependent [18]. However, ARC mostly influences the effectiveness of time-dependent renally cleared antibacterial drugs (e.g., beta-lactam antibiotics or vancomycin) [19]. In the literature, there is not a lot of data about ARC’s role in colistin therapy. There is information about other lipopolypeptide antibacterial agents with concentration-dependent killing such as daptomycin, in which the pharmacokinetic ARC does not play an important role [5]. Nevertheless, patients with ARC have a lower incidence of AKI, and our study failed to prove a protective role of ARC in the case of colistin AKI.

In this study, colistin-induced kidney injury was detected in 22% of the cases with a median onset of AKI of 7.5 days. Mostly, it was a first- or second-degree kidney injury. For example, this onset and degree of colistin AKI, corresponds with a previously published Dalfino et al. study in which median onset of colistin AKI was five days and in 42% of cases, colistin AKI was classified as first degree [17]. However, incidence of colistin-induced AKI among critically ill patients of different clinical setting varied a lot, from 10–49% [7]. Hartzell et al. demonstrated that colistin-induced AKI is usually mild and reversible [20]. In our study, no cases of permanent renal function loss or RRT due to colistin therapy were detected. However, in many cases it was not possible to evaluate due to a high death rate among the patients in the study. Generally, the incidence and severity of colistin AKI revealed in this study corresponded with previously reported data.

Baseline renal impairment and older age were detected as colistin nephrotoxicity risk factors in the study of Dalfino et al. [17]. However, this study did not show significant differences in the parameters comparing patient groups with and without colistin-induced AKI. In addition, cumulative colistin doses were not different among these two groups. This study revealed that higher baseline CRP levels were associated with higher risk of colistin-induced nephrotoxicity. Previously, higher CRP level has been reported as a risk factor for the development of radiocontrast media-induced AKI in patients undergoing percutaneous coronary intervention [21]. However, we cannot exclude the possibility that higher CRP was an indirect indicator of more severe comorbidities of the patients. Further investigation of CRP level prior to colistin therapy as a possible risk marker of colistin AKI could be beneficial.

If we take into account that the precise estimation of patient renal functional state is difficult in the routine clinical practice of intensive care units, it is possible that the use of a standard dose rather than an increased dose of colistin is safer in patients with renal hyperfiltration until TDM of colistin is available. The use of TDM could theoretically be helpful for rational colistin dosing, as some pharmacokinetic studies of colistin have already revealed inter-individual variations of colistin concentration among critically ill patients on colistin therapy-for example, Karnik et al. found that the concentration at steady state varied widely, from 0.1–2.0 mg/L [22]. Unfortunately, the target concentration of colistin is not currently clear. Usually, target antibiotic concentrations should be analyzed in the context of breakpoint MIC. For *A. baumannii*, the MIC50 is usually 0.5 mg/L and the MIC90 is 2 mg/L. For *P. aeruginosa*, it is 2/2 mg/L, and for *K. pneumoniae*, it is 1/1.5 mg/L. According to some study results, colistin nephrotoxicity could correlate with colistin concentration in the higher blood concentration (2.5 mg/L). This is associated with higher AKI risk in comparison with concentrations of 1.3–2.0 mg/L, but a possible target concentration of less than 2.5 mg/L is more likely to produce bacteriostatic rather than bactericidal effect [13].

This study has several limitations, such as its retrospective design, heterogenicity of the study population, difficulties in assessing patients’ renal functional state, and lack of information about colistin concentrations in the blood. Nevertheless, it revealed challenges of colistin therapy in real clinical situations. In addition, this study emphasized that the lack of clear recommendations from the manufacturer could result in potentially inappropriate dosing and that TDM could be potentially beneficial for some critically ill patients.

## 5. Conclusions

Colistin doses varied widely among patients, and dose adjustment was not TDM guided. In many cases, colistin doses were lower or higher than is currently recommended, especially in patients with renal impairment prior to colistin therapy. Colistin AKI risk was not correlated with serum creatinine value prior to therapy or with the colistin cumulative dose, but rather correlated with the CRP value at baseline. Patients with ARC received higher cumulative colistin doses and had a tendency towards lower incidence of colistin AKI than patients with lower GFR on baseline. It is important to evaluate the safety of higher colistin daily doses in patients with ARC in well-designed studies before recommending its use in routine clinical practice.

## Figures and Tables

**Figure 1 medicina-55-00033-f001:**
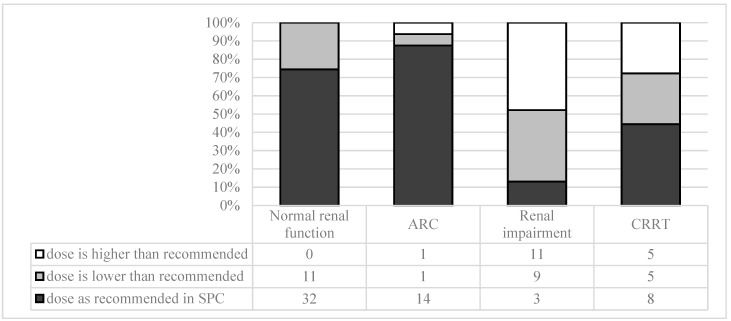
Colistin second-day dosing in comparison with manufacturer recommendations. Abbreviations: ARC: augmented renal clearance; CRRT: continuous renal replacement therapy; SPC: summary of product characteristics.

**Table 1 medicina-55-00033-t001:** Demographics, clinical diagnoses, and MDR Gram-negative infection in study population (n = 97).

Characteristics	Values
Gender: men, n (%)	65 (67%)
Age, years	
Median (Q_1_; Q_3_)	63.0 (51.0; 73.5)
Min–max	20.0–92.0
Duration of hospitalization, days	
Median (Q_1_; Q_3_)	43.0 (26.5; 63.5)
Min–max	10.0–204.0
Patient death rate, n (%)	49.0 (50.5%)
Main clinical diagnoses groups, n (%):	
Pulmonology (e.g., pneumonia, COPD)	28 (28.9%)
Neurology (e.g., subarachnoid hemorrhage)	21 (21.6%)
Cardiology (e.g., myocardial infarction, ACS)	19 (19.6%)
Gastroenterology (e.g., severe acute pancreatitis)	8 (8.2%)
Other (e.g., cancer, acute kidney failure, trauma)	21 (21.6%)
ICU where colistin therapy was started, n (%)	
General	74 (74%)
Cardiology	17 (17%)
Pulmonology	7 (7%)
Neurology	2 (2%)
Hospitalization days when MDR Gram-negative bacteria infection was diagnosed	
Median (Q_1_; Q_3_)	9.0 (12.5; 21.0)
Min–max	3.0–78.0
Source of MDR Gram-negative bacteria infection	
Trachea aspirate, n (%)	68 (68%)
Blood, n (%)	5 (5%)
Blood and trachea aspirate, n (%)	10 (10%)
Other material, including blood, n (%)	2 (2%)
Other material, without blood, n (%)	15 (15%)

Abbreviations: ACS: acute coronary syndrome; COPD: chronic obstructive lung disease; ICU: intensive care unit; MDR: multidrug-resistant.

**Table 2 medicina-55-00033-t002:** Patients’ age and colistin use in patients with different renal functional state.

	Normal Renal Function	ARC	Renal Impairment	CRRT	Statistical Significance *
Median age (Q1; Q3), years	66 (59; 76)	49 (40; 61)	65 (56; 80)	60.5 (49; 71)	0.0005
Median duration of colistin therapy (Q1; Q3), days	9 (7;17)	16 (7; 27)	10 (4; 15)	11.5 (7; 24)	0.3440
Median cumulative colistin dose (Q1; Q3), MU	78 (45; 135)	107.5 (60; 171)	72 (30; 186)	70.3 (52; 141)	0.7140

* Kruskal-Wallis test.

**Table 3 medicina-55-00033-t003:** Summary of colistin use pattern and outcomes in patients with augmented renal function (n = 16).

No	GFR Prior to Colistin Therapy	*A. baumannii* Localization	Colistin Regimen (after LD) *	Duration of Colistin Therapy, Days	Increasing of Cr >1.5× during Colistin Use	Outcome	Amount of Days from Colistin Discontinuation until Outcome
**1**	121	wound	S	40	no	alive	12
**2**	125	trachea aspirate	↑(2 days)→S	6	no	death	1
**3**	130	trachea aspirate	S	6	no	alive	28
**4**	163	trachea aspirate	S	7	no	alive	1
**5**	139	wound	S	30	no	alive	35
**6**	114	trachea aspirate	S	7	yes, but GFR >50 mL/min	alive	4
**7**	135	trachea aspirate	S	8	no	alive	0
**8**	118	blood, trachea, wound	S	15	AKI (last day of colistin use)	alive	35
**9**	114	trachea aspirate	S	7	no	death	1
**10**	119	trachea aspirate	S	3	no	alive	18
**11**	144	wound	S	17	yes, but GFR >50 mL/min	death	12
**12**	121	trachea aspirate	↓	40	no	alive	15
**13**	135	trachea aspirate	S	21	no	death	0
**14**	120	trachea aspirate	S	24	no	death	20
**15**	131	trachea aspirate	S	12	AKI (last day of colistin use)	death	0
**16**	136	trachea aspirate	S	33	no	alive	28

Abbreviations: AKI: acute kidney injury); Cr: serum creatinine; GFR: glomerular filtration rate; LD: loading dose. * Colistin regimen: S-standard (3 million units (MU) three times daily); ↑-increased (4 MU three times daily); ↓-decreased (2 MU twice daily).

**Table 4 medicina-55-00033-t004:** Association of different factors with colistin-induced acute kidney injury.

Parameter, Median (Q1; Q3)	Patients without Colistin AKI, n = 65	Patients with Colistin AKI, n = 18	Statistical Significance *
Age, years	63.0 (52.0; 74.0)	64.5 (61.0; 77.0)	0.169
Serum creatinine at the beginning of colistin therapy, µmol/L	83.0 (54.0; 121.0)	82.0 (64.0; 146.0)	0.541
CRP at the beginning of colistin therapy	87.0 (66.0; 150.0)	164.0 (119.0; 182.0)	0.012
Cumulative colistin dose until outcome **	72.0 (45.0; 141.0)	72.0 (48.0; 126.0)	0.578

* Mann-Whitney U Test. ** Outcome-stop of colistin, patient death, or colistin AKI. Abbreviations: AKI: acute kidney injury; CRP: C-reactive protein.

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
