# Peer review of "Colistin Use in Patients with Extreme Renal Function: From Dialysis to Augmented Clearance"

_medicina, 2019, doi:10.3390/medicina55020033_

Round 1
Reviewer 1 Report
Comments :
Thank you for giving me the opportunity to review the manuscript by Aitullina et colleagues about colistin use in patients with extreme renal function. The authors describe here the use of colistin in a single intensive care units, in patients with different renal functional states.
The main limitation is that it reports a single center experience.
I have few comments:
- I feel that the paper should benefit of a revision from an English native speaking reader, to enhance some of the formulations.
- Introduction line 55:
I am not sure I can understand the sentence: “In case of renal replacement therapy (RRT), there is no mechanism like tubular 55 reabsorption to return colistin from dialysate to blood and because of this colistin could be efficiently cleared during haemodialysis”
- How many patients were hospitalized more than one time during the study period?
- The author should discuss the diagnosis of ARC. Especially discuss the problem of malnutrishment, especially in patients in ICU. Creatinin being synthesized from muscles metabolism, it is well known that low creatinin levels are seen in malnutrished patients, and eGFR does not estimate accurately the real renal function. I would like to have information about the proportion of malnutrished patients, notably in the ARC group. This information could be added in table 2.
- In the specific pattern of patients with ARC, could the author estimate the proportion of failure of treatment of the infection, according to the dose of colistin administered?
- CRP was found to be associated with a greater risk of colistin induced AKI. However, this result is not adjusted on comorbidities, and there might exist a confusion bias, as patients with the higher CRP are probably sicker than others.
- The abbreviation TDM for Therapeutic Drug Monitoring should be avoided to my opinion, as it is a currently used abbreviation for tomodensytometry.
Author Response
Dear Professor,
Thank you very much for suggestions. We have tried to answer your questions and comments (attached file).
Sincerely yours,
Aleksandra Aitullina

Reviewer 2 Report
This manuscript has dealt with a relatively less attended subject and is worth encouragement. However, authors have to revise this manuscript to fit the scientific standard of medical journal. Opinions aree as following.
Reading this manuscript takes a lot of energy to catch their points, mainly because of the English. It's better to edit this manuscript by a more experienced English user to make it readable. (I have to say it's a challenge to read this one.)
Augmented renal clearane is a state less attended by nephrologist, but should be paid more attention to, at least in term of pharmacological dosing in critical care unit. There were similar papers on different antibiotics. Authors might search for then and take them for reference.
I understand the treatment policy differs a lot between units/countries, but the doses you mentioned for maintenance treatment are higher than suggested. Is there any reason?
It seems there was not a regulated protocol of treatment adjusted to the renal function in your unit. Therefore, the result of this manuscript is more closely your treatment experience, but not a well-designed strudy. Perhaps, authors can discuss how to improve the situation in the future.
There are many limitations in this manuscript. Please address them in the Discussion.
It's usually CKD-EPI, rather than CKI-EPI, in the literature.
Author Response

(The authors gave the same response as above.)

Reviewer 3 Report
General Comments: The aim of this study is to determine colistin use pattern in patient with different renal functional state, including extreme renal functions, in a single clinical center of Baltic country. The authors indicate this information is of importance as colistin is considered in their institution (tertiary adult university hospital) as a drug of last choice for treatment of multidrug-resistant (MDR) systemic gram-negative infections in critically ill patients. The study design is retrospective with a study period of three years (2015-2017). The inclusion and exclusion criteria are well defined. Data were collected by manual review of the medical records. I pressure no electronic data was available.
The authors make an important point of the presence of augmented renal clearance (ARC) in their patient population. Their concern is that if colistin is cleared too rapidly, systemic bioavailability could be reduced and theoretically lead to ineffective antibacterial therapy. The authors estimated glomerular filtration rate (eGFR) based on the CKI-EPI formula and defined normal values as 50 to 108 ml/min/1.73 m2).
Acute Kidney Injury (AKI) was defined according to the Acute Kidney Injury Network criteria as increasing of baseline creatinine level more than 1.5 times over baseline after at least 48 h of colistin therapy. First stage of AKI was defined by a creatinine level increase to 1.5-1.9 times over baseline, second stage by a serum creatinine increase to 2.0-2.9 times over baseline, and third stage by a serum creatinine increase greater than 3.0 times over baseline.
The Introduction, Materials & Methods, Results, and Discussion sections are well written. The references are up to date with 53% being less than five years old.
Specific Comments:
1. This study is retrospective in nature, so its utility is somewhat limited as patients were not screened prospectively to ensure all relevant data be collected at time of enrollment into the study. This does not negate the value of the findings but does limit the scope of the observations.
2. A total of 97 patients met the inclusion criteria. Of these, 3 had a second course of colistin for reinfection. The authors have entered these three patients twice in the data base making up to 100 incidents of colistin therapy. Adding three events to the original 97 is not a relevant increase in the N value. However, those three patients may have been impacted by the prior AKI so that their response to retreatment is not comparable to the original course. I recommend the second events be removed from analysis making N=97.
3. The prevalence of AKI possibly related to colistin was 23%, which is in the midrange of published values (8-30%). A total of 18 patients (assuming no retreatments) were on CRRT during the study. These patients were not on colistin prior to CRRT so there is no association between colistin and AKI in these patients. Did any patients in the AKI group require any type of renal replacement therapy?
4. A concern with this study is the use of the CKI-EPI formula for defining the ARC group. First, the authors do not indicate if their serum creatinine assay was standardized or not (see Ann Intern Med. 2006;145:247-254). Second, the base serum creatinine measurement is fundamental to the analysis. At what time was it obtained in relation to the study (how many days prior to entering ICU or starting colistin)? Were several measurements available to define a stable baseline? What was median number of samples used to define the baseline? In the case of ARC patients, did serum creatinine concentrations return to a normal range?
5. The authors should note in the Discussion that reference #11 states that “the gray zone, the bias, and precision values of CKD-EPI showed the limits of these formulas, which is only a tool for screening patients with ARC. In such circumstance, the CrCl should be measured formally toaccurately adjust dosage of drug eliminated by kidneys.”
6. In the Introduction the authors note that “There is hypothesis that development of ARC relates to the systemic inflammatory response syndrome (SIRS) that is observed in patients without chronic organ damage and with 44 trauma, pancreatitis, burns, autoimmune disorders, major surgical procedures, ischemia or sepsis. Table 3 reports a significant increase in CRP among patients who developed AKI. What were the CRP values in the four groups of patients studied?
7. In line 183 the reference should be #9 not #15.
8. In line 25 define SPC prior to using the abbreviation.
9. In line 45 add a corresponding reference to the end of the sentence (?#11). In line 99: shouldn’t the interquartile range be “Q1-Q4.” To this reviewer the notation (Q1-Q3) suggests an intertertile range.
10. I recommend the authors review the paper by De Lang et al. (De Lange DW. Glomerular hyperfiltration of antibiotics. Neth J Crit Care 2013;17:10-4.
11. The article by Hartzell et al. may also be of interest to the investigators Clinical Infectious Diseases 2009; 48:1724-8.
Author Response

(The authors gave the same response as above.)

Round 2
Reviewer 2 Report
Pleaso do send the article to language editing, as promised.